

# Fragmentation of two repelling Lund strings

**Cody B. Duncan[⋆] and Peter Skands**

School of Physics & Astronomy, Monash University, Clayton, VIC 3800, Australia

⋆ cody.duncan@monash.edu

## Abstract

Motivated by recent discoveries of flow-like effects in pp collisions, and noting that multiple string systems can form and hadronize simultaneously in such collisions, we develop a simple model for the repulsive interaction between two Lund strings with a positive (colour-oriented) overlap in rapidity. The model is formulated in momentum space and is based on a postulate of a constant net transverse momentum being acquired per unit of overlap along a common rapidity direction. To conserve energy, the strings shrink in the longitudinal direction, essentially converting a portion of the string invariant mass $m^2$ into $p_\perp^2$ for constant $m_\perp^2 = m^2 + p_\perp^2$ for each string. The reduction in string invariant mass implies a reduced overall multiplicity of produced hadrons; the increase in $p_\perp^2$ is local and only affects hadrons in the overlapping region. Starting from the simplest case of two symmetric and parallel strings with massless endpoints, we generalize to progressively more complicated configurations. We present an implementation of this model in the Pythia event generator and use it to illustrate the effects on hadron $p_\perp$ distributions and dihadron azimuthal correlations, contrasting it with the current version of the "shoving" model implemented in the same generator.



# 1   Introduction

Hadronization models play an essential role in the description of hadronic events in high-energy collisions, connecting the short-distance physics of quarks and gluons with the observable world of colourless (long-lived) hadrons via a dynamical process that enforces confinement. The two major models of hadronization used in proton-proton event generation are the Lund string model [1–4] and the cluster model [5–7], with the former implemented in Pythia [8–10] and Epos [11,12], and the latter in Herwig [13–15] and Sherpa [16,17].

While the Lund string model has been able to qualitatively describe a large number of hadron-level observables from $e^+e^-$ to proton-proton collisions across a wide range of CM energies (see e.g. MCPLOTS [18]), recent data in particular from the LHC experiments have highlighted some shortcomings. ALICE has shown unequivocally that strangeness production increases as a function of event multiplicity in minimum-bias event samples [19,20], while CMS discovered the near-side ridge in high-multiplicity events [21,22]. The latter has been elaborated upon in a number of studies by both ATLAS and CMS [23–27], and there are also several additional indications of strangeness enhancement e.g. in the underlying event [28–30]. Both of these phenomena are widely believed to have their roots in collective effects, but in the baseline Lund string model, each string hadronizes independently of the others (modulo effects of colour reconnections, see, e.g. [31,32]).

Several proposals have been made that can potentially explain these phenomena. Rope hadronization [33, 34] takes aligned strings in rapidity and enhances their string tensions based on a Casimir scaling argument [35,36], leading to increased strangeness production and higher average $p_\perp$ values in string breaks. Shoving, a mechanism for microscopic string-string interactions which generates transverse momentum pressure between overlapping strings, was proposed in [37,38] and showed long-range azimuthal correlations. Both the Rope and shoving model have been implemented in Pythia, Dipsy [39], and Angantyr [40]. Alternatively, the approach taken by Epos [12] invokes the notion of a critical string density beyond which a heavy-ion inspired hydrodynamic modelling takes over, which includes collective flow and thermally enhanced strangeness production. Yet a third line of argument is that colour recon-

nections (CR) can produce flow-like effects [41], essentially by creating net boosted hadronizing systems. Baryon-to-meson ratios may also be altered by CR effects [42] but would have to be supplemented by something like rope hadronization to significantly alter net strangeness fractions.

In [43], the authors studied the effects of a thermodynamic string fragmentation model, which used an exponential transverse mass spectrum instead of the usual Gaussian form for the Lund string model. Recent work on the cluster model has also tried to capture some of the collective-like effects seen by introducing baryonic clusters and strangeness enhancement in Herwig [44,45]. Other approaches to studying and modeling azimuthal correlations in proton-proton collision environments include the String Percolation model [46–50], or interference effects from multiple parton interactions [51] or from the BFKL parton shower evolution process [52]. Recent work has investigated the initial state geometry of the collision and the resultant effect on azimuthal anisotropy [53–55]. The Colour Glass Condesate (CGC), a successful framework for describing the collision environment in heavy-ion collisions, has also been applied to proton-proton collisions [56,57]. Kinetic transport theory has also been used to study the potential source of angular correlations [58]. A review of collectivity in small systems can be found in e.g. [59,60].

We here take the same basic starting point as the shoving model [37, 38], namely that nearby Lund strings should exert a force upon one another. We focus on repulsive forces since we assume that colour-reconnection models such as [42] (based on colour algebra and string-length minimisation) provide a first approximation to any attractive effects. We further assume that all of the hadronizing colour charges emanate from a region that is small compared with the typical width of a string. This restricts the applicability of our model to small systems but allows us the simplification of working entirely in momentum space. By contrast, the shoving model adopts an explicit picture of the spatial distribution and time evolution of the strings. (The space-time structure of hadronization in the Lund model was also recently further explored in [61].) Furthermore, the effect of the interaction is in our model represented via a global rescaling of the 4-momenta of the string endpoints combined with a local addition of $p_\perp$ to hadrons formed in regions of string overlap, while the shoving model imparts transverse momentum by adding a number of low-energy slightly massive gluons to each string. Despite similar physical starting points, we therefore do expect some qualitative differences to arise between the shoving model [37,38] and our momentum-space realization of repelling strings.

The article is organized as follows. Sec. 2 presents a short review of the Lund string model with emphasis on those features that are most relevant to our toy extension model. Sec. 3 introduces our string-string interaction model in the context of the simplest two-string configuration, and presents how the repulsion is implemented during string fragmentation, and the effects on primary hadron transverse momentum. We then extend this formalism to a more general parallel two-string configuration in Sec. 4 and then to strings with endpoints with both longitudinal and transverse momentum in Sec. 5. To make a connection with the phenomenological characterisations of collective flow used in heavy-ion inspired studies, we illustrate the effects on two-particle cumulants, $c_2\{2\}$, for selected two-string configurations in Sec. 6. In Sec. 7, we discuss the effects of decays of short-lived primary hadrons. Modifications for strings with massive endpoints are briefly discussed in Sec. 8 before we conclude and give an outlook for future work in Sec. 9.

## 2  Lund String Model

The Lund string model [1–4] is based on the linear nature of the confinement potential $V(r) = \kappa r$ between static quark-antiquark pairs separated by distances greater than about

a femtometre (see e.g. [36]). Strings are implemented in Pythia at the end of the perturbative shower, where long colour-chains produced by the shower are collected into colour singlets, the so-called Lund strings.

A Lund string represents a confined gluonic flux tube or vortex line. In the simplest case it runs between a quark endpoint via any number of intermediate gluons (which generate transverse kinks in the structure) to an antiquark endpoint. Other colour topologies are possible as well, such as junctions and gluon loops. In this work, we restrict our attention to simple $q\bar{q}$ strings without any transverse gluon excitations.

As the endpoints propagate outward in opposite directions from the production point, their energy and momentum gets transferred to the Lund string that stretches between them. When sufficient energy is available, new $q'\bar{q}'$ pairs can be produced in the string field (typically by invoking a Schwinger-type tunneling mechanism [62]); the string thereby breaks into successively shorter pieces each of which ultimately becomes an on-shell hadron, in a process called fragmentation. In the ordinary Lund string model, each string fragments independently, and each string break is independent of any others.

Fragmentation proceeds by successively splitting off one hadron from either endpoint (chosen at random), with the created hadron at each step taking a fraction $z$ of the string's available lightcone momentum distributed according to the Lund symmetric fragmentation function:

$$f(z) = N \frac{(1-z)^a}{z} \exp\left(\frac{-bm_\perp^2}{z}\right), \tag{1}$$

with the leftover string retaining the remainder $1-z$. $N$ is a normalisation constant and $a$ and $b$ are phenomenological parameters to be determined from fits to data, see e.g. [63,64]. $m_\perp^2 = m^2 + p_\perp^2$ is the transverse mass of the produced hadron; its $p_\perp$ is obtained as the (vector) sum of the $p_\perp$ values of each of its constituent quarks. In the absence of collective effects each string break is assumed to impart an equal and oppositely oriented $p_\perp$ to the produced quark and antiquark, which by default is given a Gaussian distribution, by analogy with the Schwinger mechanism in QED [62]. In the Rope model [33], the coherent fragmentation of multiple nearby colour charges can cause the width of this $p_\perp$ distribution (as well as strangeness and baryon production probabilities) to increase. While we believe those arguments to be fundamentally correct, for simplicity we focus in this work solely on the collective repulsion aspect, keeping other string-breaking aspects unmodified.

## 2.1 Fragmentation and rapidity

In the context of interacting strings, we will be interested in the effective overlap in rapidity between a produced hadron and a nearby string piece. To start with, we need an expression for the rapidity span taken by each hadron along an axis defined by its own string system.

Letting $m_0$ denote a generic hadron mass, the rapidity span of a simple $q\bar{q}$ string with massless endpoints traveling in opposite directions along the $z$-axis is:

$$\Delta y_0 = \ln\left(\frac{W_{+q}}{m_0}\right) - \left[-\ln\left(\frac{W_{-\bar{q}}}{m_0}\right)\right] = \ln\left(\frac{W^2}{m_0^2}\right), \tag{2}$$

where $W_\pm = E \pm p_z$ are their lightcone momenta, and $W^2 = W_+ W_-$ is the squared invariant mass of the string. Throughout this work, we will use the $z$ axis as the (common) rapidity axis, and our example configurations will be defined so that this is reasonable, but there is obviously nothing special about this choice; the formalism we develop can be applied for any choice of axis.

After a hadron, $h$, is split off from one of the endpoints, let the invariant mass of the leftover string be $W'^2$. The size of the rapidity interval associated with the produced hadron can then

be identified with the difference:

$$\Delta y_h = \ln\left(\frac{W^2}{m_0^2}\right) - \ln\left(\frac{W'^2}{m_0^2}\right) = \ln\left(\frac{W^2}{W'^2}\right) , \tag{3}$$

which is independent of $m_0$. App. A elaborates on how Eq. (3) relates to the sequence of $z$ fractions and hadron mass values for arbitrary (sequences of) string breaks, using the notation from [4, 61] which also matches the code implementation. Below, we shall use these expressions to quantify the total rapidity overlap that a given hadron has with a nearby string piece.

## 3 Repulsion Between Two Parallel Identical Strings

We start by considering the simplest possible configuration: two straight and parallel strings of the same squared invariant mass, $W^2$.

Viewed in space-time, the repulsion between two such strings should depend on their (time-dependent) transverse separation distance [65, 66]. However, in the context of hadronization in high-energy particle collisions, the preceding perturbative stages of event generation are normally treated in momentum space, i.e. in terms of plane-wave approximations that are not well localized in space-time. Thus, one faces a problem of mapping partons represented in momentum space onto string systems represented in space-time. In the framework of classical string theory, on which the Lund model is based, one may simply use the string tension $\kappa$ to convert between the two pictures. But when multiple string systems are involved, any interactions between them will depend on the space-time separation between the production points of each system, which the momentum-space perturbative boundary conditions only serve to fix up to an ambiguity $\propto 1/\Lambda_{\rm QCD}$. Moreover, while a strict classical interpretation would in principle allow for arbitrarily small separations, string descriptions are only appropriate for long-distance QCD. Interesting work has been done recently to bridge the two pictures [61, 67], but for the purpose of this study we would like to explore how far we can get if we stay in momentum space.

Our underlying assumption will be that our colliding systems are of order a hadronic size (hence we do not address heavy ions) and that, by the time strings are formed, they are already at least some "typical" transverse distance apart, again of order hadronic sizes even if the directions of motion of the endpoints were originally completely parallel. We make the boost-invariant ansatz that parallel strings impart a constant amount of net transverse momentum to each other per unit of overlap in rapidity,

$$\frac{\mathrm{d}p_{\perp R}}{dy} = c_R , \tag{4}$$

where the constant $c_R$, which has dimensions of GeV per unit rapidity, represents the main tuneable parameter in our model. It controls the strength of the repulsion, or alternatively, the conversion strength of longitudinal momentum into transverse momentum.[1] Non-parallel configurations will be discussed below. We further make the ansatz that each hadron produced in the overlap region receives a fraction of the total repulsion $p_\perp$ in proportion to (the overlapping portion of) its rapidity span according to Eq. 3.

A schematic diagram of how our model works is shown in Fig. 1. In a first step, we remove an amount of longitudinal momentum from the original endpoints ("compression"),

---

[1]In a future extension we shall relate this to an increase in the tension of the individual strings as well, in a manner similar to what is done in Rope hadronization, but this is outside the scope of this work.

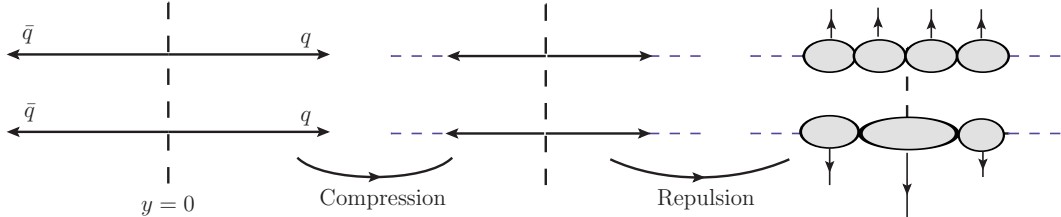

Figure 1: Schematic diagram of the simplest two-string configuration and the two steps in our model: compressing the strings (black solid lines) and repulsion during the string's fragmentation. The hadrons (grey ovals) receive $p_\perp$ proportional to the string length they take. We have ignored the Gaussian transverse momentum generation in the Lund string model for the purposes of this figure. The transverse separation between the strings in the diagram is for clarity.

in proportion to the size of the total rapidity overlap between the two strings. In the second step, the energy that was removed in the compression step is imparted back to the hadrons formed in the region(s) of overlap, as transverse momentum ("repulsion").

### 3.1 String Compression

Each string is defined by its two endpoints, which for simplicity we take to be massless for now and travelling in opposite directions along the $z$ axis. Right-moving endpoints thus start out with lightcone momenta $W_+ = 2E$ and $W_- = 0$ and vice versa for the left-moving ones. In the fully symmetric setup we consider here, both strings will undergo the same transformations described below. We focus on just one of them.

Since the strings have equal invariant masses, the overlap is simply the full rapidity span of each string, i.e. $\Delta y_{\rm ov} = \Delta y_{\rm string}$, which is given by Eq. (2). For this work, we found that using too small an $m_0$ can lead to pathological results since this presumes that every hadron you can create has an invariant mass of that order. Instead, we will choose to work with $m_0 = m_\rho = 0.77$ GeV. Thus, by integration of Eq. (4) the $p_\perp$ gained by each string will be:

$$p_{\perp R} = \pm c_R \cdot \Delta y_{\rm ov}, \tag{5}$$

where the $\pm$ sign symbolically represents that the kicks will act in opposite directions, so that no net $p_\perp$ is gained by the string-string system as a whole.

To conserve energy, this $p_\perp$ must be acquired at the expense of some amount of longitudinal momentum. We start by defining a set of intermediate rescaled lightcone momenta $W'_\pm = f_\pm W_\pm$ with

$$f_+ f_- = 1 - \frac{p_{\perp,R}^2}{W^2} \le 1, \tag{6}$$

which corresponds to a $W'$ string system with a lower invariant mass,

$$W'_- W'_+ = W'^2 = W^2 - p_{\perp R}^2 . \tag{7}$$

This first step of the model is illustrated by the left-hand part of Fig. 1, labelled "Compression". In the simple case studied in this section the compression factors $f_+$ and $f_-$ must be equal for symmetry reasons. (More general cases, with $f_+ \neq f_-$, will be considered in the next section.)

A particularly simple way of representing the repulsion effect would be to boost the $W'$ system transversely by a factor $\vec{\beta}_\perp = \vec{p}_{\perp R}/W'$. However, as G. Gustafson demonstrated during enjoyable discussions in Lund, such a boost would assign relatively more of the repulsion $p_\perp$

to high-rapidity hadrons than to central ones, in contrast with the manifestly longitudinally invariant form of Eq. (4). Instead, we therefore modify the fragmentation of the $W'$ system in a more local way, by allowing each produced hadron to receive an additional amount of $p_\perp$ in a manner designed to reproduce Eq. (4).

Writing the 4-vectors as $(p_+, p_-, \vec{p}_\perp)$, the $W'$ system is defined by:

$$
\begin{aligned}
p'_q &= f W_+ \left(1, 0, \vec{0}_\perp\right), \\
p'_{\bar{q}} &= f W_- \left(0, 1, \vec{0}_\perp\right) .
\end{aligned}
\tag{8}
$$

As remarked above, this has a lower total energy, $W'$, than that of the original system. The "missing energy" will gradually be added back during the fragmentation process, in the form of additional $p_\perp$ given to the hadrons that are formed in the region(s) of overlap. Unlike the standard fragmentation $p_\perp$ in string breaks, which is randomly and independently distributed in azimuth for each breakup, a single global $\phi$ choice characterises the $p_\perp$ component from repulsion (with $\pi + \phi$ used for the hadrons in the recoiling string system). We will now discuss the details of this second step, illustrated by the right-hand part of Fig. 1, labelled "Repulsion".

## 3.2 Repulsion

As mentioned in Sec. 2.1, we can assign a rapidity span to each hadron as it gets produced by the rapidity span lost by the string when producing the hadron. Using Eq. (36), a hadron receives a corresponding fraction of $p_{\perp R}$, calculated in the same manner as Eq. (5):

$$
p_{\perp h} = c_R \Delta y_h = p_{\perp R} \frac{\Delta y_h}{\Delta y_{\text{string}}},
\tag{9}
$$

where $\Delta y_h$ is the rapidity span of the string taken by the hadron, such that $\sum \Delta y_h = \Delta y_{\text{string}}$, and consequently the summed repulsion momentum given to hadrons is equal to the total repulsion momentum. Generalising to cases in which the two strings do not fully overlap, the numerator and denominator of the rapidity-span ratio in the last expression can simply be changed to refer to the overlapping portions of the hadron and total rapidity spans, respectively. After the hadron receives the repulsion $p_\perp$, its energy is then adjusted by the amount required to put it back on shell. In this way, the "missing energy" discussed above is gradually added back to the system.

Note that, if there were no other sources of transverse momentum, putting a hadron on-shell after the repulsion would always increase its energy. However, since each string break is associated with a randomly distributed fragmentation $p_\perp$ (with each hadron in general receiving contributions from two such breaks), which must be added vectorially to the repulsion $p_\perp$, some hadrons may have lower total $p_\perp$ after adding the repulsion effect. In our modeling setup, such hadrons are regarded as donating some energy back to the string system's reservoir of "missing energy", with the sum over all hadrons still respecting eq. (5).

With this modification, we follow the same iterative fragmentation procedure as in ordinary Pythia, splitting off hadrons from either end, allowing them to receive additional repulsion $p_\perp$ and putting them back on shell, until the invariant mass of the remaining string system drops below a cutoff value:

$$
W_{\text{rem}}^2 < W_{\text{stop}}^2.
\tag{10}
$$

At this point, we add any remaining repulsion $p_\perp$ to the remnant object, as well as any energy that is still missing from the compression process. This makes total energy and momentum conservation explicit. Pythia then produces two final hadrons from this modified remnant string.

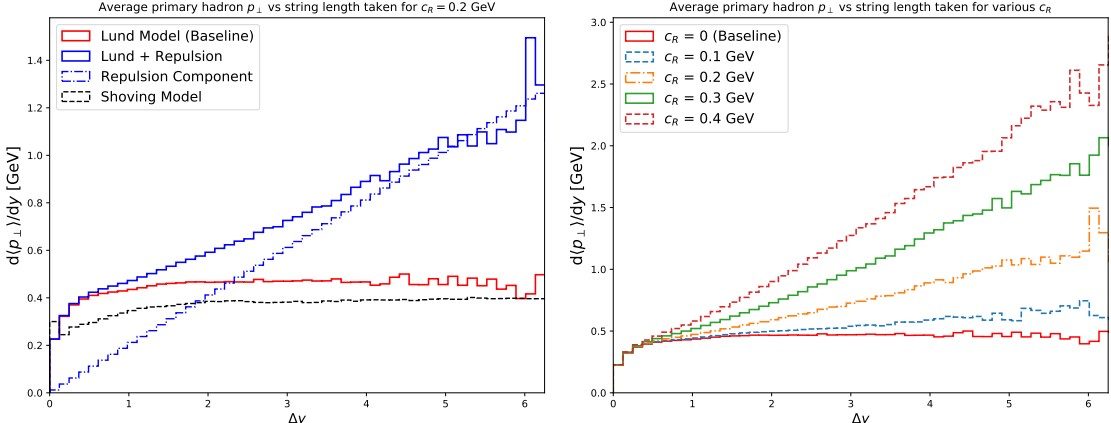

Figure 2: Distribution of average primary-hadron $p_\perp$ as a function of $\Delta y_h$. *Left:* comparison of baseline Lund model (red solid line) to our model for $c_R = 0.2$ GeV (blue solid line), our model with only the repulsion $p_\perp$ component (blue dot-dashed line) and the shoving model (black dashed line). The shoving model exhibits a lower average $p_\perp$ since the soft gluons it adds make the strings longer causing the multiplicity of produced hadrons to increase faster than the total $p_\perp$. *Right:* the effect of varying the repulsion strength $c_R$.

## 3.3 Results

In the rest of this section, we study the consequences of our model for an explicit example configuration defined by:

$$
\begin{aligned}
p_{+1} &= p_{+2} = 400\left(1, 0, \vec{0}_\perp\right) \text{GeV} , \\
p_{-1} &= p_{-2} = 400\left(0, 1, \vec{0}_\perp\right) \text{GeV} .
\end{aligned}
\tag{11}
$$

To highlight the effects of the fragmentation repulsion, we have chosen endpoint energies of 200 GeV (corresponding to rather long strings), and, at this stage, consider only primary hadrons (hadrons that are produced directly from the fragmenting string). The smearing caused by decays of (short-lived) primary hadrons into secondaries will be discussed in Sec. 7.

The left pane of Fig. 2 shows the average $p_\perp$ of primary hadrons as a function of $\Delta y_h$, as defined by Eq. (3). The red dashed histogram shows the results of using the ordinary Lund model, which — since the Gaussian transverse momentum generation in the baseline Lund model is independent of the rapidity span — is a flat distribution modulo endpoint effects, The two blue histograms illustrate the effects of our compression and fragmentation repulsion model, for a representative value of $c_R = 0.2$ GeV. The dot-dashed histogram shows the repulsion component by itself (obtained by turning off the Gaussian fragmentation $p_\perp$ component via `StringPT:sigma = 0`). The solid blue histogram shows the combination of the fragmentation and repulsion $p_\perp$ components, for the same reference value of $c_R$. For small $\Delta y$, this mimics the baseline string model, while for large $\Delta y$, the repulsion $p_\perp$ takes over as the dominant source of transverse momentum.

We also include a comparison to the shoving model as implemented in Pythia 8.2 [37,38]. For the shoving parameters used in our study (see App. B for details), the average transverse momentum per unit rapidity span taken actually decreases relative to the baseline (solid red) model. We interpret this as a result of the physical mechanism by which the shoving model pushes the two strings apart, which is implemented as a number of very soft transverse gluon excitations. While this does increase the total $p_\perp$, it also increases the total string length. The latter in turn increases the hadron multiplicity, with the result that the average $p_\perp$ per

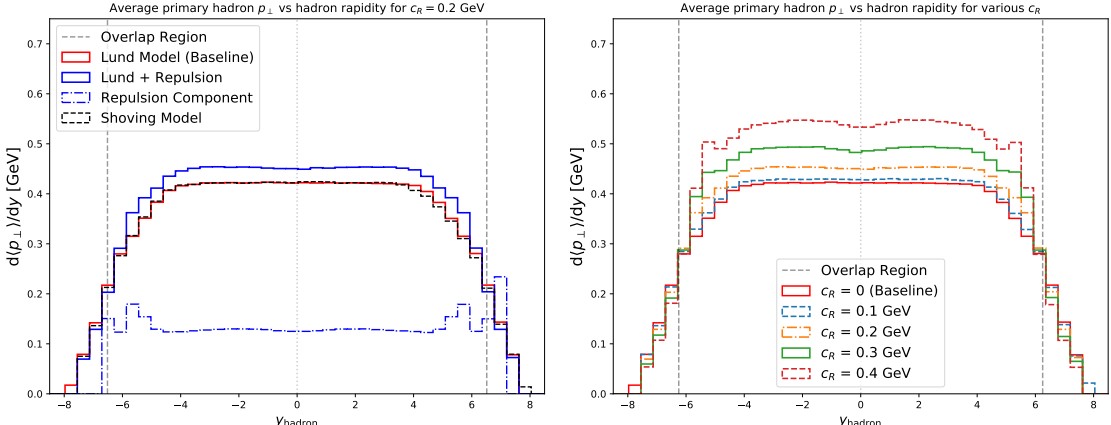

Figure 3: Distributions for the average $p_\perp$ of primary hadrons as a function of the hadron's rapidity for the symmetric parallel strings configuration. *Left*: comparison of the baseline Lund model (red solid line), with our fragmentation repulsion model (blue solid line), which has a higher $\langle p_\perp \rangle$ in the plateau region. The component which is due to the repulsion effect is illustrated by the blue dot-dashed line. Also shown is the result of using the shoving model (black dashed line) [38], for the same string configuration. The shoving model does not have significant deviation from the baseline Lund model for this observable (see text). *Right*: the effect that varying the repulsion strength $c_R$.

hadron can decrease. In our model, by contrast, the compression step ensures that the total multiplicity decreases; the repulsion step then adds $p_\perp$, implying that both the total and the average $p_\perp$ per hadron must increase.

The results of varying $c_R$ from 0 GeV (equivalent to the no-repulsion baseline case) to 0.4 GeV per unit of rapidity overlap are shown in the right panel of Fig. 2. As $c_R$ increases, the slope of the average hadron $p_\perp$ increases with the rapidity span of the string taken, as expected from the ansatz in Eq. (9).

In Fig. 3, we show the same model examples but now as a function of the more directly observable rapidity of the hadrons, instead of the rapidity span they take. For the normal Lund string model, this produces a variant of the famous rapidity plateau (red solid line). For the parameters we studied, the shoving model (dashed black line) does not change the average $p_\perp$ appreciably (while the average multiplicity of the event is increased [38]). In contrast, for our reference value of $c = 0.2$ GeV, our repulsion model (blue solid line, with the repulsion component illustrated by the blue dashed line) does increase the average primary hadron $p_\perp$. The net increase is less than linear since the ordinary (Gaussian) fragmentation $p_\perp$ is oriented randomly with respect to the repulsion $p_\perp$, and the two components add vectorially.

As in the previous figure, the right panel of Fig. 3 illustrates the effect of varying $c_R$ in the range 0 to 0.4 GeV per unit of rapidity overlap. For larger values of $c_R$, the rapidity plateau begins to lose some of its flat structure, particularly in the middle of the string, near $y_{\text{hadron}} = 0$. To fix the flatness, one may adjust the stopping mass parameter $W_{\text{stop}}^2$ in Pythia's implementation of the string model, though this is outside the scope of this work.

## 4  General Parallel Two-String Configuration

We now extend the considerations in Sec. 3 to a more general configuration, by letting the strings have an arbitrary parallel configuration. Without loss of generality, we assume that

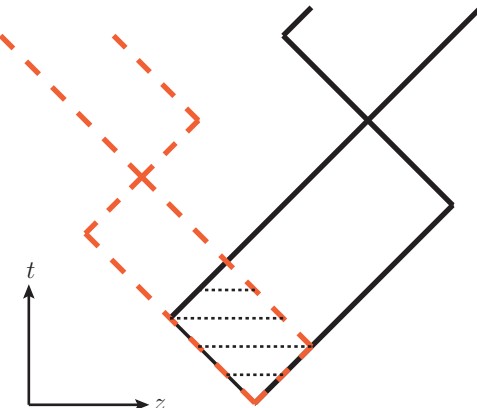

Figure 4: Schematic (1+1)-D spacetime diagram of the general parallel two-string configuration, where the two strings have a region of overlap (dotted parallel lines). The two endpoints in the region of overlap will be subjected to more compression and repulsion.

the two strings do still overlap, either partially, or one string's rapidity span is fully contained inside the rapidity span of the other. Relabeling as needed, we require in the former case that the left-moving ($W_-$) end of string 1 is contained within the rapidity span of string 2, and the right-moving ($W_+$) end of string 2 is contained within the rapidity span of string 1.

In the context of the momentum-space representation of the Lund model that our repulsion framework is based on, the full space-time evolution of a string is determined solely by the starting values of the 4-momenta of its endpoints. By initially reducing these momenta, the "compression" step of our model expresses the physical expectation that, as two nearby strings expand simultaneously and repel each other, it will not be possible to convert *all* of the kinetic energy of their endpoints into potential energy stored in the corresponding strings; instead, some fraction of the original kinetic energy is "held in reserve", to be converted into transverse momentum during the fragmentation process. When we now turn to consider asymmetric configurations, we must answer not only how much of the total kinetic energy must be held in reserve in this way, but also which fraction of it to take from each of the reservoirs represented by the two endpoints.

In our fragmentation repulsion model, we will use the ansatz that endpoints "inside" a region of overlap should undergo more compression than ones "outside", since the corresponding string regions experience more of the accumulated interaction. In Fig. 4, we show a (1+1)-D diagram of a general string configuration, with an overlapping region centred around a slightly negative rapidity (in the given frame). The right-moving endpoint of the dashed-orange string piece overlaps with the solid-black string system during the entire time over which its original kinetic energy is converted to potential energy. By contrast, the left-moving endpoint of the same dashed-orange string piece only overlaps with the black system during half of the time that it takes to convert all of its kinetic energy to potential energy. In this sense, the right-moving endpoint can be considered to be "inside" the region of overlap while the left-moving one ultimately travels "outside" of that region. Alternatively, the portion of the black-solid string system that is represented by its left-moving endpoint has a bigger fraction of total overlapping area than the portion that is represented by its right-moving endpoint.

## 4.1 String Compression

In the general case that the strings are not symmetric in the longitudinal direction, one must make a choice whether to allow them to exchange $p_L$ or not. For simplicity and since we

wish to focus on the transverse repulsion effects here, we choose to ignore the possibility of $p_L$ exchange in this first version version of our model. Thus, the only change with respect to the symmetric case is that the rescaling factors for each of the four endpoint momenta will no longer be equal.

Regardless of longitudinal recoil, the compression factors for each string system $i \in [1,2]$ must satisfy:

$$
\begin{aligned}
f_{+1}f_{-1} = f_1^2 &= 1 - \frac{p_{\perp,R}^2}{W_1^2} , \\
f_{+2}f_{-2} = f_2^2 &= 1 - \frac{p_{\perp,R}^2}{W_2^2} ,
\end{aligned}
\tag{12}
$$

where $p_{\perp,R}$ is the total $p_\perp \propto \Delta y_{\rm ov}$ from repulsion to be assigned (equally and oppositely) to the two systems, see eq. (5), and longitudinal momentum conservation, $\Delta p_{L,1} = -\Delta p_{L,2}$, implies:

$$
(1-f_{+1})W_{+1} - (1-f_{-1})W_{-1} = (1-f_{-2})W_{-2} - (1-f_{+2})W_{+2} . \tag{13}
$$

This gives three constraints for four unknowns. Imposing the further condition of no longitudinal momentum exchange, $\Delta p_{L,1} = \Delta p_{L,2} = 0$, eq. (13) separates into:

$$
\begin{aligned}
(1-f_{+1})W_{+1} - (1-f_{-1})W_{-1} &= 0, \\
(1-f_{-2})W_{-2} - (1-f_{+2})W_{+2} &= 0.
\end{aligned}
\tag{14}
$$

The problem can then be solved with a unique set of solutions for each compression factor $f_{\pm i}$. Inserting the first two constraints Eq. (12) into Eq. (14), we obtain a quadratic equation for $f_{-i}$:

$$
W_{-i}f_{-i}^2 + (W_{+i} - W_{-i})f_{-i} - f_i^2 W_{+i} = 0. \tag{15}
$$

Since the compression factors must be positive, there is only one solution to this equation:

$$
f_{-i} = \frac{(W_{-i} - W_{+i}) + \sqrt{(W_{-i} - W_{+i})^2 + 4W_i^2 f_i^2}}{2W_{-i}}, \tag{16}
$$

or equivalently using the longitudinal momentum component $W_{Li} = (W_{+i} - W_{-i})/2$,

$$
\begin{aligned}
W_{-i}' = f_{-i}W_{-i} &= \sqrt{W_{Li}^2 + W_i^2 f_i^2} - W_{Li} , \\
W_{+i}' = f_{+i}W_{+i} &= \sqrt{W_{Li}^2 + W_i^2 f_i^2} + W_{Li} .
\end{aligned}
\tag{17}
$$

In the limit of $W_{+i} = W_{-i}$, i.e. $W_{Li} = 0$, we reproduce the symmetric case for the given string $i$, i.e. $f_{\pm i} = \sqrt{f_i^2}$. By construction, longitudinal momentum is conserved, $W_{+i}' - W_{-i}' = W_{+i} - W_{-i}$. However, energy is not:

$$
E_i' = \frac{W_{+i}' + W_{-i}'}{2} = E_i \sqrt{1 - \frac{p_{\perp,R}^2}{E_i^2}} . \tag{18}
$$

When we perform the fragmentation repulsion, we regain the "lost" energy by giving the primary hadrons the repulsion $p_\perp$ and putting them on-shell again, with the string remnant absorbing the remaining energy. Thus, we conserve energy and momentum after compression *and* fragmentation of the strings.

It should be mentioned that our choice of no $p_L$ exchange does introduce a dependence on the frame in which the system is considered. This is due to the fact that while the lightcone momenta $W_\pm$ follow a simple rescaling under longitudinal boosts, the compression factors



Figure 5: Schematic diagram of the general two-string configuration and how we perform the string compression using Eq. (16), and then the fragmentation repulsion. Only primary hadrons in the region of overlap will receive $p_\perp$ proportional to the string length taken. We have ignored the Gaussian transverse momentum generation in the Lund string model for the purposes of this figure.

$f_{\pm i}$ depend non-linearly on $W_{\pm i}$ as seen in Eq. (16), complicating their transformations under such boosts. Specifically, compressing the strings then boosting the entire system results in a (marginally) different momentum topology than boosting the strings with the same boost factor and then compressing them. In this work unless otherwise stated, we compute compression factors in the overall CM frame of the two-string system. (A possible alternative, not pursued here, would be to boost the system longitudinally such that the centre of the overlap region is at $y = 0$.)

## 4.2 Repulsion

The repulsion effect we seek to model is local; additional $p_\perp$ should be imparted to hadrons formed within regions of string overlap, and not to those outside. Fragmenting the (compressed) string from the outside in as usual, and using Eq. (3) to compute rapidity spans, we distinguish three cases for each produced hadron:

1. The span is completely outside the overlap region;

2. The span is completely inside the overlap region;

3. The span straddles the boundary of the overlap region.

In the first case, the hadron receives no repulsion $p_\perp$, while in the second, it is computed according to Eq. (9) and assigned repulsion $p_\perp$ following the same procedures as described in Sec. 3. In the last case, only the portion of the rapidity span inside the overlap region contributes to Eq. (9).

To illustrate the repulsion effect we consider a two-string scenario defined by the following endpoints (using the same lightcone notation as previously),

$$
\begin{aligned}
p_{+1} &= 1200\left(1, 0, \vec{0}_\perp\right) \text{ GeV}, \\
p_{-1} &= \phantom{0}300\left(0, 1, \vec{0}_\perp\right) \text{ GeV}, \\
p_{+2} &= \phantom{0}100\left(1, 0, \vec{0}_\perp\right) \text{ GeV}, \\
p_{-2} &= 1000\left(0, 1, \vec{0}_\perp\right) \text{ GeV},
\end{aligned}
\tag{19}
$$

This configuration is then boosted back to the overall CM frame. An illustration of the compression and repulsion steps for this type of configuration is given in Fig. 5.

## 4.3 Results

In Fig. 6, we show the average primary hadron $p_\perp$ distribution as a function of the string rapidity span taken by the hadron.

In the left panel of Fig. 6, the red histogram is the ordinary Lund model, which is agnostic to the the rapidity span taken by a hadron. The blue histograms are the result of our implemented

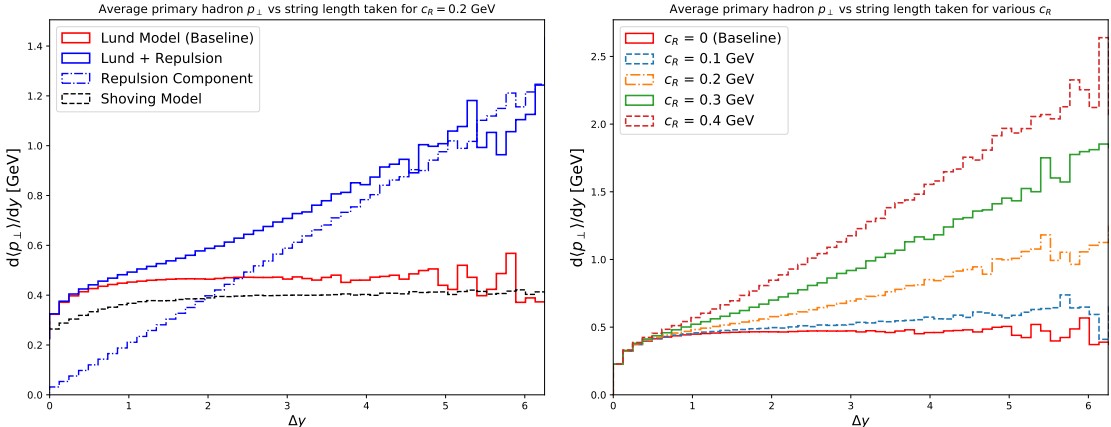

Figure 6: Distribution of average hadron $p_\perp$ for primary hadrons as a function of the rapidity span that they take, for the asymmetric two-string configuration discussed in the text. *Left*: the baseline Lund model (red solid) compared to our model of fragmentation repulsion (blue solid line). For the latter, the blue dot-dashed histogram illustrates the component which is due to repulsion. Also shown is the result of using the shoving model, which, like the baseline Lund model, is agnostic to the amount of string length taken. *Right*: the effect of varying $c_R$ in Eq. (9).

model for $c_R = 0.2$ GeV, for both the repulsion component (dot-dashed), which matches the ansatz in Eq. (9), and the full fragmentation (solid), which matches the baseline Lund model and the repulsion component in the limits of small and large $\Delta y$ respectively. Lastly, we have also included the results of using the shoving model for this configuration. These results are largely similar to the results for the symmetric, parallel configuration in Sec. 3.3.

The right panel of Fig. 6 highlights the effects of varying $c_R$ on the average primary hadron $p_\perp$ distribution as a function of the string's rapidity span taken by the hadron for the full fragmentation repulsion. For $c_R = 0$ GeV, we reproduce the ordinary Lund model. As the repulsion factor increases, the slope of the average $p_\perp$ increases, since the two are proportional via Eq. (9).

In Fig. 7, we present the average primary hadron $p_\perp$ distribution as a function of the hadron's rapidity (as measured in the overall CM frame of the two strings). Since the configuration is asymmetric with respect to the endpoints of the two strings, the resultant compression and fragmentation repulsion will also reflect this asymmetry. The red histogram is the ordinary Lund string model, and again we reproduce the rapidity plateau, with a small asymmetry due to the configuration of strings. The blue histograms are our fragmentation repulsion for $c_R = 0.2$ GeV where we have shown only the repulsion component (dot-dashed), and the full fragmentation repulsion (solid). We have also included the results of the shoving model (black dashed).

Fig. 7 also showcases the considerations from Sec. 4.2. In comparison to Fig. 3 where the repulsion component has a sharp cut-off at the edges of the rapidity overlap region, in the general case we have longer tails that extend beyond the overlap region due to hadrons taking rapidity spans that are only partially in the overlap region.

Comparing Figs. 3 and 7, we see the same structures for each respective model. Our fragmentation repulsion exhibits an increased average $p_\perp$ for hadrons inside the rapidity overlap region, while hadrons outside that region have a diminished $p_\perp$ contribution from the repulsion. As in the previous section, we see that the shoving model considered in this study does not change the distribution, apart from minor deviations near the endpoints.

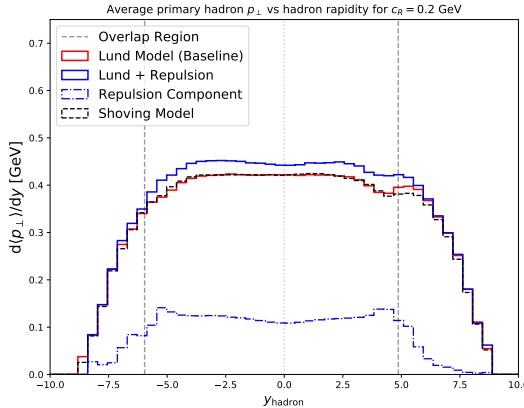

Figure 7: Distribution of average hadron $p_\perp$ for primary hadrons as a function of $y_{\text{hadron}}$, for the asymmetric two-string example described in the text. The repulsion component of our fragmentation repulsion increases the $\langle p_\perp \rangle$ in the region of overlap (indicated by the grey dashed lines, using $m_0 = 0.5$ GeV in the rapidity calculation).

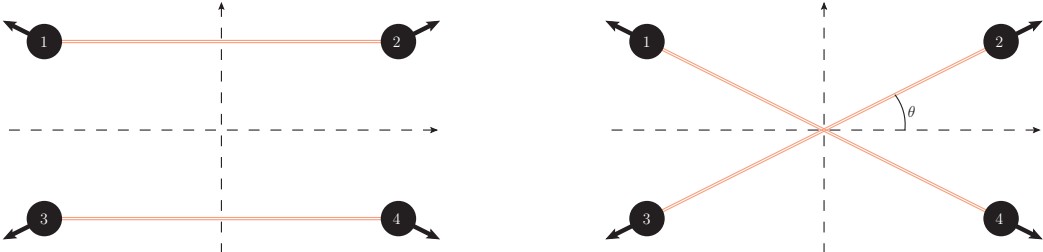

Figure 8: Schematic diagram of two string systems defined by the endpoint momenta given in Eq. (20), corresponding to (left) a relative boost and (right) a relative rotation.

## 5 Two-String Systems with Relative Rotations and Boosts

We now consider string systems with endpoints that have non-vanishing transverse momenta. The examples we consider in this section will still be defined so that the $z$ axis remains a sensible choice of common rapidity axis. Specifically, we will consider systems like those illustrated in Fig. 8, with endpoint momenta (in conventional 4-momentum notation):

$$
\begin{aligned}
p_1 &= E(\ 1, & \sin\theta, & \quad 0, & -\cos\theta\ ), \\
p_2 &= E(\ 1, & \sin\theta, & \quad 0, & \cos\theta\ ), \\
p_3 &= E(\ 1, & -\sin\theta, & \quad 0, & -\cos\theta\ ), \\
p_4 &= E(\ 1, & -\sin\theta, & \quad 0, & \cos\theta\ ),
\end{aligned}
\tag{20}
$$

so that the string systems defined by the (1,2) and (3,4) pairings are still parallel but each are transversely boosted relative to the overall CM, by $\beta = \pm \sin\theta$, while the systems defined by the pairings (1,4) and (2,3) are at rest relative to the overall CM but are rotated with respect to each other, with a relative opening angle of $2\theta$. In all cases, the CM energy is $E_{\text{CM}} = 4E$. For definiteness we take $\sin\theta = 0.1$ in the examples below unless otherwise stated.

### 5.1 Symmetric configuration with relative boost

Taking the simplest symmetric two-string configuration, we ask what happens in the situation depicted in the left-hand panel of Fig. 8 in which both strings have some (equal and opposite)

transverse momentum before compression.

Using the same arguments as above, we wish to convert a fraction of the original longitudinal momenta of the endpoints (defined along the common rapidity axis, here the $z$ axis) into transverse momentum instead of into potential energy of the string(s).

As before, the total amount of repulsion $p_\perp$ is determined from the effective rapidity overlap, which we compute from the longitudinal momentum components (along the chosen common axis) of the endpoints,

$$\Delta y_{\rm ov} = \min(y_{1+}, y_{2+}) - \max(y_{1-}, y_{2-}) , \tag{21}$$

where $y_{i+}$ and $y_{i-}$ refer to the rapidities of the right- and left-moving endpoints of string $i$ respectively and we regulate the rapidity values of massless endpoints in the $p_\perp \to 0$ limit by imposing $m \geq m_0$ in the denominator of our rapidity definition:

$$y = \ln \frac{E + p_L}{\sqrt{m^2 + p_\perp^2}} . \tag{22}$$

Using lightcone coordinates as before, the longitudinal component of a general string-end momentum is $p_L = (W_+ - W_-)/2$, and the energy is $E = (W_+ + W_-)/2$.

The string-ends will be rescaled in a similar manner to the parallel strings in Sec. 3. Since the rescaling is done on the full 4-vectors, the string endpoints will lose some $p_\perp$. We use the ansatz of giving this extra transverse momentum reservoir, denoted $p_{\perp,{\rm res}}$ to the fragmenting hadrons as a fraction of the rapidity span they take from the string:

$$p_{\perp,{\rm h}} = \left( c_R + \frac{p_{\perp,{\rm res}}}{\Delta y_{\rm ov}} \right) \Delta y_h, \tag{23}$$

where $\Delta y_{\rm ov}$ is string-string overlap defined via Eq. (21, and $\Delta y_h$ is the amount of rapidity span taken by the hadron inside of the overlap region, as discussed in Sec. 4. (Alternatively, and probably more correctly, one could distribute $p_{\perp,{\rm res}}$ among all the hadrons, not just those in the overlap region; or boosting the compressed string transversely so that it regains its original total $p_\perp$; but since since $p_{\perp,{\rm res}}$ is typically very small it is a minor effect.)

As in the previous section we assume no longitudinal momentum exchange, $\Delta p_{\rm L} = 0$. Writing the total longitudinal momentum of string $i \in [1,2]$ as

$$p_{L,i} = p_{L,+i} + p_{L,-i} , \tag{24}$$

with $p_{L,\pm i}$ the longitudinal momentum of the respective endpoints, we can generalize Eq. (16) to:

$$f_{-i} = \frac{p_{L,i} + \sqrt{p_{L,i}^2 - 4 p_{L,-i} p_{L,+i} f_i^2}}{2 p_{L,-i}}. \tag{25}$$

In the limit of the string ends carrying $p_\perp \to 0$, Eq. (25) exactly reproduces Eq. (16).

The amount of repulsion $\perp$ given to each hadron during the fragmentation process should be proportional to the (overlapping portion of the) rapidity span it takes. The definition, Eq. (36), is given in terms of the quantities used to characterize the fragmentation of each string in its own CM frame, along the axis defined by its endpoints in that frame, whereas we here want to along the chosen common axis in the string-string CM frame. As a very simple way to "project" the rapidity span, we use

$$\Delta y_{\rm eff} = \frac{\Delta y_{\rm string}}{\Delta y_{\rm string}^*} \Delta y_{\rm taken}^* , \tag{26}$$



where the $\Delta y_{\text{string}}$ is the rapidity span of the given string evaluated along the common axis defined in the string-string frame and $\Delta y_{\text{string}}^* = \ln\left(W^2/m_0^2\right)$ is the (larger) span evaluated in the string's own rest frame. $\Delta y_{\text{taken}}^* = \ln\left(W^2/W'^2\right)$ is the rapidity span of the hadron taken in the string's own rest frame.

The effective string length in Eq. (26) taken is invariant under longitudinal boosts, and reproduces the parallel configuration in the limit where each string endpoint carries vanishing $p_\perp$. Eq. (26) also sums to give the correct rapidity span along the $z$-axis, and is agnostic to the direction of the transverse momentum.

The last point to address is in which direction in azimuth to apply the repulsion. Considering the transverse plane only (in the string-string CM frame), the two systems will have some equal and opposite overall motion, which we denote by $\vec{p}_{\perp,\text{rel}} = \vec{p}_{\perp 1} - \vec{p}_{\perp 2} = 2\vec{p}_{\perp 1}$. Assuming that, by the time strings are formed, the string systems are already separated a bit (on average) along this axis, it seems plausible to us to apply the repulsion $p_\perp$ along the same direction. To provide some variability and in order to have a well-defined repulsion axis also in the $p_{\perp,\text{rel}}\,to0$ limit, we add a random component as well:

$$\vec{n}_{\perp 1} = N(\vec{p}_{\perp,\text{rel}} + \rho\vec{n}_{\perp,\text{ran}}),\tag{27}$$

where $\vec{n}_{\perp,\text{ran}}$ is a unit-vector in a randomly chosen azimuthal direction, the normalisation factor

$$N = \frac{1}{\sqrt{p_{\perp,\text{rel}}^2 + \rho^2 + 2\rho(\vec{p}_{\perp,\text{rel}} \cdot \vec{n}_{\perp,\text{ran}})}}\tag{28}$$

ensures $|n_{\perp 1}| = 1$, and $\rho$ is a free parameter of order 1 GeV which governs the relative importance of the random component. The repulsion for string 1 is oriented *with* $n_{\perp 1}$, and that for string 2 in the opposite direction.

The choice of direction can have a significant effect on two-particle azimuthal correlations, as we will describe in Sec. 6, but it does not have a drastic effect at the level of the distributions for the average hadron transverse momentum versus hadron rapidity and rapidity span taken.

## 5.2 Results

We present the results in Fig. 9. Both panels show the average primary hadron $p_\perp$ as a function of $y_{\text{hadron}}$ (defined along the common string-string axis, here the $z$ axis, in their overall CM frame). In these plots we have chosen to add the repulsion $p_\perp$ in the same direction as each string's overall transverse motion, and chosen a larger value of $c_R = 0.4$ GeV compared to the parallel configurations, to make the effects of the repulsion stand out a bit more clearly against the $p_\perp$ contributed already from the endpoints.

In Fig. 9, our fragmentation repulsion exhibits similar effects as those seen in Figs. 3 and 7, though the enhancement of average primary hadron $p_\perp$ is less drastic than in the parallel configurations. The reason for this is twofold: first, since the strings are no longer parallel, the amount of rapidity overlap between the two strings is reduced, resulting in less total repulsion $p_\perp$. Second, the boosted endpoints show up as peaked structures around the endpoints' rapidity.

While the effects of our fragmentation repulsion are less distinctive in Fig. 9, the framework will still have a distinctive effect on the two-particle azimuthal correlations, as we will discuss in Sec. 6.

## 5.3 Asymmetric configurations

Generalizing to an arbitrary configuration with strings that have endpoints with transverse momentum follows naturally from combining the frameworks presented in Sec. 4 and Sec. 5.1.

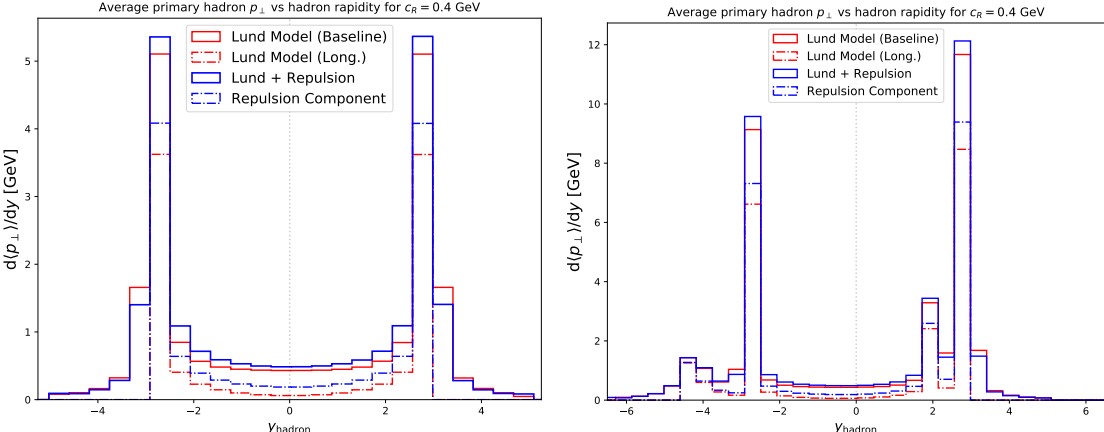

Figure 9: Distribution of average hadron $p_\perp$ of primary hadrons as a function of the hadron's rapidity, for the symmetric configuration (left) and the general configuration (right), where the two strings have an equal and opposite boost in the transverse direction. The latter configuration is boosted back to the two-string rest frame before compression and fragmentation. We have added the repulsion $p_\perp$ in the same direction as the overall motion of each string.

The effects are smaller than in the symmetric configuration with opposite boosts, since the overlap in rapidity along the $z$-axis decreases the more transversely boosted the endpoints are. This results in less compression, and less fragmentation repulsion. We will use the configuration from Eq. (19), with a boost factor of $\beta = 0.1$ in opposite directions for each string.

In the right panel of 9, we show the results of boosting each string in the general configuration given by Eq. (19) in opposite directions, then boosting back to their common rest frame, and then performing our compression and fragmentation repulsion. We have chosen to present the results of using $c_R = 0.4$ GeV since larger values of this parameter are required to have visible results for this observable. The results are in line with our expectations from previous sections, namely that strings with endpoints that have transverse components will compress and repel less than strings that are completely parallel, and similarly with strings that are not completely overlapping.

## 5.4 Rotated configurations

Configurations such as those depicted in the right-hand pane of Fig. 8 can be treated using the same arguments as for the boosted configurations. The endpoints again have non-vanishing transverse momenta, hence the rapidity spans computed along the common rapidity axis are always smaller than those in the respective string CM frames.

In the specific example shown in 8, $p_{\perp,\text{res}} = 0$ since each of the (1,4) and (2,3) strings have zero net $p_\perp$. Compression factors are computed from the longitudinal momentum components as in Eq. (25, and the effective span taken by each hadron is projected onto the common axis using Eq. (26).

Finally, since each of the strings are at rest the $\vec{p}_{\perp,\text{rel}}$ in Eq. (27) is zero hence the random component will dominate in the choice of azimuth direction. (A more physical choice could potentially be made by using the direction transverse to the plane spanned by the two strings, but since we consider the case of vanishing $\vec{p}_{\perp,\text{rel}}$ to be of limited general interest we do not pursue this further here.)

There are many other configurations that one may consider, but with the four configurations discussed in this work, we have presented the overall framework for our model of

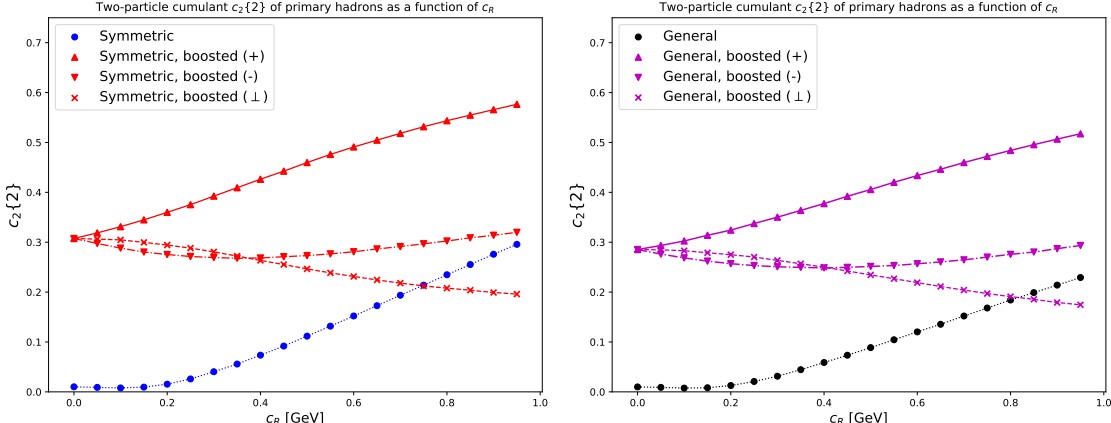

Figure 10: Two-particle cumulant for the symmetric (left panel) and the general (right panel) two-string configurations, at the level of primary hadron production. We show the curves for the simplest parallel two-string case, and three variations on the equal and oppositely boosted two-string case. The variations are: the repulsion $p_\perp$ acts in the same direction as the given string's overall transverse motion ("Boosted, (+)"), the repulsion $p_\perp$ acts in the opposite direction ("Boosted, (−)"), and lastly, the repulsion $p_\perp$ acts perpendicularly to the string's boost ("Boosted, (⊥)"). For each curve, when $c_R = 0$, we reproduce the baseline Lund string model.

fragmentation repulsion.

## 6 Flow and Cumulants for Two-String Configurations

Long-distance correlations in rapidity and azimuth have been used extensively to probe collective aspects of event structure, including flow, in both proton-proton and heavy-ion collisions. (See, e.g., [68] for a succinct review of elliptic flow in heavy-ion phenomenology, and references therein.) Here, we focus on just one such observable, the two-particle cumulant, $c_2\{2\}$, which is designed to suppress non-flow contributions. It is calculated as:

$$c_2\{2\} = \left\langle \langle e^{2i(\phi_i - \phi_j)} \rangle \right\rangle = \left\langle \frac{2}{n(n-1)} \sum_{i<j}^{n} \cos\left(2(\phi_i - \phi_j)\right) \right\rangle, \tag{29}$$

where in the first line the outer angle bracket is the average over all events, and the inner is the average over all $n$ particles in a given event. In the second line of Eq. (29), we have removed the self-correlations $i = j$, and used the fact that the cosine function is an even function.

The two-particle cumulant will depend not only on the repulsion strength $c_R$, but also on the direction of the repulsion, in particular for cases where the strings have an overall transverse motion such as the transversely boosted strings, where $\vec{v}_{\text{det}} \neq 0$. In this work, we will simply show the three extreme cases of the repulsion directions for the transversely boosted configurations, as discussed in Sec. 5.2

In Fig. 10, we plot the results for the two-particle cumulant for the symmetric two-string configuration at the level of primary hadrons, as a function of the repulsion constant $c_R$. There are four curves in the plot. The first curve, labelled 'Symmetric' is the simplest two-string configuration, considered in Sec. 3. In this configuration, there is no preferred $\phi$ direction, and it takes larger values of the repulsion constant to overcome the Gaussian transverse momentum distribution of the Lund fragmentation model, and to have a significant effect on the cumulant.

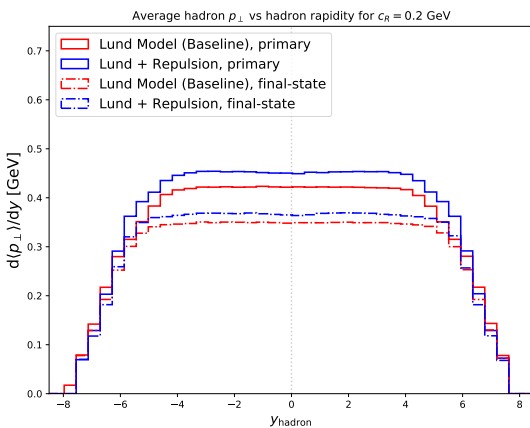

Figure 11: Illustration of the net reduction of average hadron $p_\perp$ caused by allowing excited primary hadrons (solid histograms) to decay (dot-dashed histograms), for the baseline Lund model (red) and our fragmentation repulsion model (blue). The example configuration is the symmetric parallel two-string configuration described in Sec. 3; the primary-hadron spectra are the same as those in Fig. 3.

The three other curves are variations on the configuration described in Sec. 5.1 where the two strings each have a boost of $\beta = 0.1$ in equal and opposite directions. The variations occur when one adds the repulsion $p_\perp$ to the primary hadrons during fragmentation. The curves are labelled according to the direction in which the repulsion $p_\perp$ is added with respect to the given string's overall boost direction. If we add the repulsion $p_\perp$ in the same direction as the string's motion, we can greatly enhance the two-particle cumulant. If instead we add it in the opposite direction, we at first reduce the two-particle cumulant, but as the repulsion gets larger, the cumulant begins to increase. Lastly, if we add the repulsion $p_\perp$ perpendicularly to the string's motion we greatly reduce the cumulant, but at large values of the repulsion constant, the rate of decrease begins to level out.

We obtain analogous results for the general configuration in the right panel of Fig. 10, though the cumulant for all values of $c_R$ is less than for the symmetric case, due to the smaller overlap in rapidity.

We compared the symmetric parallel configuration in our fragmentation repulsion framework to the analogous configuration in the shoving model, and found that the two-particle cumulant is significantly smaller for the shoving model, at least with the parameter set described in App. B. For the shoving model, we calculated the two-particle cumulant to be $c_2\{2\} = 0.00957$ (averaged over 200,000 events), which is of the order of the baseline Lund model.

# 7 Final-State Hadrons

In the previous sections, we considered the $p_\perp$ and rapidity distributions at the level of the primary hadrons produced in the fragmentation process. Decays of those hadrons into secondaries (via processes like $\rho \to \pi\pi$, $\pi^0 \to \gamma\gamma$, etc.) will smear the distributions in rapidity and dilute the $p_\perp$ enhancement per hadron. In this section, we include decays of all final-state particles with lifetimes shorter than $\tau = 10$ mm/$c$. In Pythia, this is done with the two switches: `ParticleDecays:limitTau0 = on`, and `ParticleDecays:tau0Max = 10`. With this criterion, weakly decaying strange hadrons are treated as stable, while all particles with shorter lifetimes are decayed. This matches the typical definition for stable par-

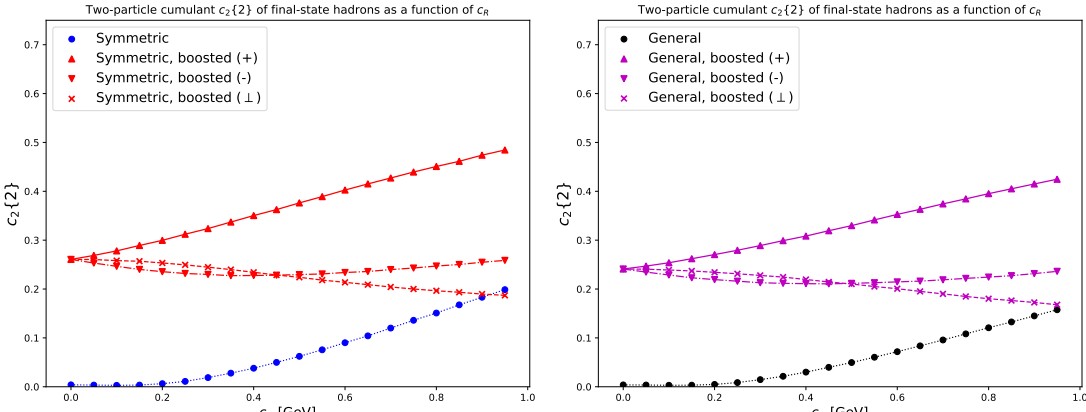

Figure 12: Two-particle cumulant for final-state hadrons in the symmetric configuration (left), and the general configuration (right), as a function of the repulsion constant $c_R$. Both plots exhibit the same trends as the primary hadron distributions in Fig. 10, though the correlations are slightly reduced, as expected from excited hadrons decaying isotropically into potentially non-hadronic final states.

ticles used at LHC.

In Fig. 11, we present the average hadron $p_\perp$ distribution as a function of hadron rapidity, for the symmetric parallel configuration from Sec. 3. We replot the results for the baseline Lund model (red solid) and our fragmentation repulsion (blue solid) for primary hadrons. Allowing excited primary hadrons to decay produces the dot-dashed lines in Fig. 11, for the baseline Lund (red dot-dashed) and our fragmentation repulsion (blue dot-dashed).

As expected, the plateau has been lowered for the baseline Lund model, since excited primary hadrons can decay into non-hadronic final state particles, which remove some of the available $p_\perp$. Similarly, the fragmentation repulsion exhibits a lowering of its peak and general structure. However, the difference between the structure of the fragmentation repulsion and the rapidity plateau of the Lund model remains intact when decays are turned on, meaning our model can still be distinguished from the baseline Lund model.

In Fig. 12, we show the effects of varying the repulsion constant $c_R$ on the two-particle azimuthal cumulant $c_2\{2\}$ of final-state hadrons, for the symmetric configurations (left) and the general configurations (right). As shown, the cumulant exhibits the same trends as the primary hadron counterparts in Fig. 10, though the effects have been somewhat reduced, due to the non-hadronic particles produced during particle decays.

The key result of allowing particle decays is that our fragmentation repulsion model, implemented at the level of the primary hadrons produced during string fragmentation, still retains its key signatures at the level of final-state hadrons, at least at the level of the two-string configurations.

## 8 Strings With Massive Endpoints

The final generalisation we will consider in this work concerns strings with massive endpoints. The starting point for the compression process is the same as in the massless case, in that we rescale the 4-momenta as if the endpoints were massless:

$$p_\pm^\mu \to p_\pm'^\mu = f_\pm p_\pm^\mu, \tag{30}$$

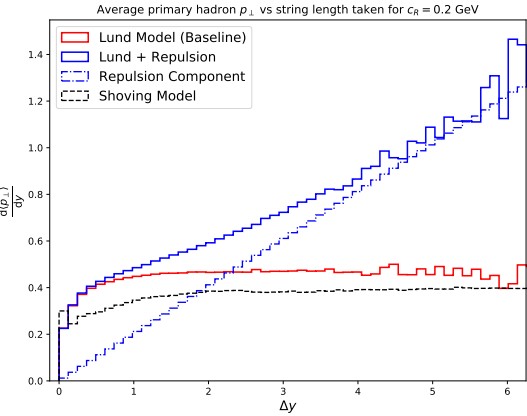

Figure 13: Distribution of average hadron $p_\perp$ for primary hadrons as a function of the rapidity span of the string taken by the hadron, for the symmetric, parallel two-string configuration with massive endpoints.

where the subscript $\pm$ refers to the positively and negatively $z$-aligned endpoints respectively. The compression factors are, however, slightly modified relative to those in Eq. (12). Using the conservation of invariant mass:

$$W'^2 = W^2 - p_{\perp,R}^2,$$
$$\text{thus } (f_+ p_+ + f_- p_-)^2 = (p_+ + p_-)^2 - p_{\perp,R}^2, \tag{31}$$

where we have inserted the original and rescaled endpoint momenta in the second line.

Expanding Eq. (31) and rearranging gives:

$$(1 - f_+^2)m_+^2 + (1 - f_-^2)m_-^2 + 2p_+ \cdot p_- - p_{\perp,R}^2 = 2f_+ f_- p_+ \cdot p_-. \tag{32}$$

Using the longitudinal momentum conservation to remove, e.g., $f_+$ produces a quadratic in $f_-$ which can be simply solved to calculate the two compression factors.

After calculating the new momenta for the endpoints in the manner described above, we put the endpoints back on shell:

$$E'_\pm = \sqrt{m_\pm^2 + \vec{p}'^2_\pm} = \sqrt{m_\pm^2 + f_\pm^2 \vec{p}_\pm^2} \le E_\pm, \tag{33}$$

where the last inequality of Eq. (33) emphasises the fact that $f_\pm$ are indeed compression factors. With Eqs. (32, 33), we now have a prescription for compressing strings with massive endpoints. The repulsion part is the same as that described in Secs. 3.2 and 5.3.

In Fig. 13, we present the results of our fragmentation repulsion model for the symmetric, parallel two-string configuration with massive endpoints. As expected, Fig. 13 reproduces the same characteristics as Fig. 2, and in particular, the significant difference between the shoving model and the Lund model with our fragmentation repulsion remains.

Lastly, with the above prescription for handling symmetric, parallel strings with massive endpoints, we can extend this formalism to the general two-string configuration using the frameworks of this section and Sec. 5.3. A full presentation of this and an extension to strings with gluon kinks will be discussed in future work.

## 9 Conclusion and Outlook

We have presented a framework to compress two simple $q\bar{q}$ strings and repel them at the level of string fragmentation, a model we call fragmentation repulsion. We have shown that this

induces an increased average $p_\perp$ per hadron in regions of string overlaps and that this in turn generates non-trivial two-particle azimuthal correlations.

With the configurations presented, one may begin to build up the more complicated string topologies from the smaller pieces we have considered. Future work will look first at strings with gluon kinks, then at configurations with more than two overlapping strings. More complicated string topologies such as junctions and closed gluon loops will also need to be addressed to turn the model into a full-fledged description of LHC events.

A shortcoming of our work is that it does not provide a microscopic description of the string-string interactions, unlike the shoving model. That is, we describe the effect simply in terms of an effective average $p_\perp$ density that we postulate is accumulated by strings that overlap in rapidity, and which is transferred to the hadrons that are produced in the overlapping regions. Despite its relative simplicity, the model exhibits distinctive signatures in both average hadron transverse momentum and two-particle azimuthal correlations which are easy to understand intuitively. The amount of repulsion generated via Eq. (5) is longitudinally boost invariant, but there remains some frame dependence — and associated ambiguities — in our choices of rapidity and repulsion axes, and in the definition of the compression procedure. We aim to study these aspects further in future work.

We round off by noting that, since the cluster hadronization model is based on simple $q\bar{q}$ systems not unlike those considered here, it might be possible to apply our model also in the context of the cluster model, to let clusters repel off one another while losing some longitudinal momentum. However, since a cluster undergoes fissioning and decay, the repulsion would need to be split between the two products in the respective processes.

# Acknowledgements

We thank Gösta Gustafson for his valuable comments on the work, Christian Bierlich for his helpful comments about the current implementation of the shoving model in Pythia 8.2, and Helen Brooks & Johannes Bellm for discussions of our work. CBD is supported by the Australian Government Research Training Program Scholarship and the J. L. William Scholarship. This work was funded in part by the Australian Research Council via Discovery Project DP170100708 – "Emergent Phenomena in Quantum Chromodynamics" – and in part by the European Union's Horizon 2020 research and innovation programme under the Marie Skłodowska-Curie grant agreement No 722105 – MCnetITN3.

# A  String Fragmentation in Pythia

The Lund string fragments probabilistically by taking steps along the lightcone momenta $W_{\pm}$, where $W_+ W_- = W^2$. A hadron is created by taking a fraction $z_h$ of a given end's lightcone momentum and the rest of the string keeps $1 - z_h$ of the lightcone momentum. In order to put the hadron on shell, we also need to take some lightcone momentum from the other end.

Following the notational convention of [4], if we have taken $i$ iterative steps in the fragmentation process, producing $q_i \bar{q}_i$ pairs, each of which take a fraction $z_i$, and $0 \leq z_i \leq 1$, we can write the fractions of the initial total $W_{\pm}$ taken at each step:

$$
x_{+,i} = z_i \prod_{j=1}^{i-1} \left(1 - z_j\right),
$$
$$
\text{and } x_{-,i} = \frac{m_{\perp,i}^2}{x_{+,i} W^2}, \tag{34}
$$
$$
\text{since } m_{\perp,i}^2 = x_{+,i} x_{-,i} W^2,
$$

where we have assumed without loss of generality that the hadrons have been fragmenting from the $W_+$ end of the string.

Since the string can fragment from either end of the string, Pythia needs two sets of these $x_{\pm}$ pairs, where now the $\pm$ sign refers to the lightcone momenta of the opposite end of the given fragmenting end. We will label them $x$ and $\tilde{x}$. These two pairs track how much has been taken from the two end points in the two different directions, and the differences are the amount of lightcone momentum actually left:

$$
\bar{x}_{\text{tot},+} = x_+ - \tilde{x}_-, \text{and } \bar{x}_{\text{tot},-} = \tilde{x}_+ - x_-. \tag{35}
$$

Using Eq. (3), we can now calculate the rapidity span of the string that a fragmenting hadron $i$ takes with it:

$$
\Delta y = \ln \left( \frac{\bar{x}_{\text{tot},+} \bar{x}_{\text{tot},-}}{(\bar{x}_{\text{tot},+} - x_{h,+})(\bar{x}_{\text{tot},-} - x_{h,-})} \right), \tag{36}
$$

where $x_{h,\pm}$ is the lightcone momentum fraction taken by a new hadron fragmentation from the positive end and negative end respectively.

At some cutoff invariant mass $W_{\text{stop}}^2$, this fragmentation process stops, and the remnant string is broken into two final hadrons.

# B  Shoving Model Parameters

In the shoving model (as implemented in Pythia 8.2), there are several parameters that govern the rate and amount of shoving. We summarise the parameter values we used to produce Fig. 3 in Tab. 1. We did not include the flavour changing aspects of the Rope model.

We also set the two strings' endpoints to have $m_u = 0.33$ GeV, though this configuration and our massless endpoint configuration were set to have the same total invariant mass for each string. Since the shoving model also requires partons to have transverse spacetime coordinates, we set the strings to be 2.46 fm apart in transverse space (six times the input parameter Ropewalk:r0). We chose to set the strings relatively far apart, relative to the transverse radius of the string, since we discovered that for the above parameter set, a transverse separation between our two straight strings of $d_{\perp} < 5r_0$ lead to, in our opinion, pathological results. To understand what each parameter governs in the model, we direct the reader to [38].

Table 1: Input parameters used in Fig. 3 for the shoving model.

| Parameter | Value |
|---|---|
| Ropewalk:rCutOff | 10.0 |
| Ropewalk:limitMom | on |
| Ropewalk:pTcut | 2.0 |
| Ropewalk:r0 | 0.41 |
| Ropewalk:m0 | 0.2 |
| Ropewalk:gAmplitude | 10.0 |
| Ropewalk:gExponent | 1.0 |
| Ropewalk:deltat | 0.1 |
| Ropewalk:tShove | 1.0 |
| Ropewalk:deltay | 0.1 |
| Ropewalk:tInit | 1.5 |

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
