# Peer review of "Fragmentation of Two Repelling Lund Strings"

_SciPost Physics, doi:SciPost Phys. 8, 080 (2020)_

## Round 1 · Referee Report · Anonymous (Referee 1) · 2020-2-16

Strengths

The paper is very clear and easy to follow.

Weaknesses

1) The paper is based completely on ad-hoc assumptions, which are such that they lead obviously to the desired result. For example, the "symmetric boosted" case leads obviously to non-zero c2{2}, based on the fact that the four string endpoints are very much correlated. Why should it be so? I think this is an extremely unlikely situation. The authors admit "A shortcoming of our work is that it does not provide a microscopic description...". It is not really necessary to always provide a microscopic description, but even a toy model should at least provide some (good) arguments why the assumptions are reasonable. 2)Also the term "QCD strings" in the title is missleading, at no point "QCD" enters in the discussion of the model.

Report

Report on Fragmentation of Two Repelling QCD Strings
by Cody B Duncan and Peter Skands

The authors present a simple model for the repulsive interaction between two strings with an overlap in rapidity, which has the effect of increasing pt and reducing multiplicity. The model is motivated by recent discoveries of flow-like effects in pp scattering.

The model is first discussed based on two parallel Lund strings. The model is formulated in momentum space (although in principle space-time should play a role). The main ingredient is the assumption of a constant pt transfer per rapidity from one string to the other. At the same time, the strings get shorter in rapidity. For the new strings, the modified fragmentation takes into account locally the transferred pt, such that each hadron covering a rapidity range Delta_y gets a fraction Delta_y / Delta_y(string) of the pt transfer. In 3.3 some results of a simple example are shown (which are kind of obvious).

In chapter 4 the model is generalized to two partly overlapping strings, such that pt transfer happens to hadrons formed within regions of string overlap. Again results of a simple example are shown (which again are kind of obvious). In chapter 5 finally one considers two strings with endpoints moving also in transverse direction. In the "symmetric boosted" case, the two end partons in each string move in the same direction. In chapters 6 and 7 effects of decay and finite end point masses are discussed.

In chapter 6, results concerning flow (in terms of c2{2}) are discussed, probably meant to be one of the key results of the paper. For the "symmetric" case (purely longitudinal), a sizable repulsion constant c_R is needed to get non-zero c2{2}, in the "symmetric boosted" case, already c_R = 0 gives non-zero c2{2}.

The paper is very clear and easy to follow, BUT it is based completely on ad-hoc assumptions, which are such that they lead obviously to the desired result. For example, the "symmetric boosted" case leads obviously to non-zero c2{2}, based on the fact that the four string endpoints are very much correlated. Why should it be so? I think this is an extremely unlikely situation. The authors admit "A shortcoming of our work is that it does not provide a microscopic description...". It is not really necessary to always provide a microscopic description, but even a toy model should at least provide some (good) arguments why the assumptions are reasonable. Also the term "QCD strings" in the title is missleading, at no point "QCD" enters in the discussion of the model.

Requested changes

see "Weaknesses", in particular point 1). I expect a serious discussion and a solid justification (and not just some superficial discussion)

  • validity: -
  • significance: -
  • originality: -
  • clarity: -
  • formatting: -
  • grammar: -

Author:  Cody B Duncan  on 2020-04-22  [id 803]

(in reply to Report 1 on 2020-02-16)
Category:
remark
answer to question

We thank the referee for their careful reading of the manuscript and their helpful comments. We note that we have updated the title of the paper to be ‘Fragmentation of Two Repelling Lund Strings’ to reflect the referees’ collective comment on the misuse of the previous phrase ‘QCD Strings’, as well as cleaned up several references to ‘QCD strings’ in the introduction. We have also included more references to work done in a similar vein but from different approaches. The corresponding statement in the introduction of our paper that reads as:

'Other approaches to studying and modeling azimuthal correlation sin proton-proton collision environments include the String Percolation model [46–50], or interference effects from multiple parton interactions [51] or from the BFKL parton shower evolution process [52]. Recent work has investigated the initial state geometry of the collision and the resultant effect on azimuthal anisotropy [53–55]. The Colour Glass Condesate (CGC), a successful framework for describing the collision environment in heavy-ion collisions, has also been applied to proton-proton collisions [56, 57]. Kinetic transport theory has also been used to study the potential source of angular correlations [58]. A review of collectivity in small systems can be found in e.g. [59, 60].'

It is unclear what notion the Referee refers to as ‘ad-hoc’ in the paper. If it is the ansatz that the transverse repulsion from overlapping strings is constant per unit of rapidity, this ansatz is well-motivated by the fact that we are dealing with Lund strings. These strings must have a string-like behaviour. In the original Lund string model, the string tension per unit ‘length’ must be constant, and the multiplicity produced from a string per unit rapidity is also a constant. Accordingly, the transverse momentum donated and received by each string should scale linearly with some measure of the string’s length, the most natural measure of which (in the Lund string model) is rapidity span. (This also makes the formalism Lorentz invariant with respect to boosts along the chosen rapidity axis.)

If the 4 different configurations are unclear as physically motivating scenarios, we would note that the manuscript is meant to outline the model in a step-by-step manner. - Correspondingly, we start with the simplest case, the symmetric parallel configuration, a scenario that is unlikely to occur in a real collision, but acts the baseline for our model, and lays out the bones of how we repel the two strings. - We then transition to the general parallel configuration, a scenario that is more likely to occur in a real collision, and use this transition to note that one needs more nuance to describe the repulsion between the two strings. In particular, the two strings do not fully overlap, and as a result we do not allow the strings ends outside of the overlap region to receive any transverse momentum. - In the third case, we extend the formalism to deal with endpoints with transverse momentum, taking one more step to a more physical scenario of two strings. Granted, the symmetry of this configuration makes it an unlikely occurrence in a collision. - In the fourth case, we finally put the pieces of the second and third configurations to describe a physical configuration of two strings, one that is general enough to describe the interaction between any two simple Lund strings (with massless endpoints, though this gets extended in the final section).

While configurations 1 and 3 are unlikely to occur in a real collision, 2 might, and 4 is highly likely, given that we design it to be a general description. One can always boost to the rest frame of two strings to get a configuration akin to the general configuration with transverse motion.

Anonymous on 2020-04-28  [id 811]

(in reply to Cody B Duncan on 2020-04-22 [id 803])

Certainly useful to have removed "QCD" and added references.

What remains unclear is the physics picture behind the repulsion in general (not so much the question of a constant pt transfer per rapidity unit). Before discovering v2 in pp, the idea was more that strings simply fuse.

The other point is the role of the endpoints. It seems that even in the "generalized case" the endpoint momenta are highly correlated, which makes a large fraction of the effect concerning c2. Such correlations should be very unlikely in reality, where should they come from?

---

## Round 1 · Referee Report · Anonymous (Referee 2) · 2020-2-26

Report

Dear Editor the paper deals with the modification of the classical string Lund model, in which the authors include the repulsion of the two strings in the momentum space. The authors study the repulsion mechanism in detail for different two string configurations. The authors claim that the repulsion will lead to effective decrease of the quark mass and consequently to the increased strange production that is indeed observed experimentally. Moreover the inter string interactions will lead to to nontrivial azimuthal correlations, which is similar to other similar models of such azimuthal correlations, not based on hydrodynamics. The paper is very interesting and should be published. Nevertheless it will bye worthwhile to make some improvements in the text. 1. The authors make intuitive claim that the repulsion leads to the reduction of the effective mass m that is confirmed then numerically. Since this is the basic idea of the paper it will bye very interesting if the authors will explain this effect in more detail, may be in a separate chapter and using some toy model 2. it is not clear from the paper, although there is some mentioning by authors how their results are influenced by multiplicity. Indeed we expect the success of the classical Pythia program without repulsion for conventional final states and new effects to appear for the large multiplicity events. 3. The ideas of the paper about the flow are actually conceptually similar to also to other models, not based on string, and not related to hydrodynamics that are used to study the collective flow.

  • validity: -
  • significance: -
  • originality: -
  • clarity: -
  • formatting: -
  • grammar: -

Author:  Cody B Duncan  on 2020-04-22  [id 804]

(in reply to Report 2 on 2020-02-26)
Category:
correction

We thank the Referee for their positive comments on the manuscript, and valuable points.

With regards to the Referee’s first point, it is unclear to us if they are referring to a reduction of the effective quark masses as such, or if they are referring to the reduction of the total invariant mass of the string. The former is not a feature of our model, while the latter is. Not being sure which point the referee wants us to address, we address both below.

Firstly, concerning the reduction of the overall invariant mass of the string. Each string (consisting of two endpoints) has a total invariant mass, its CM energy. At time t = 0, this consists entirely of the kinetic energy of the endpoints, which, in the baseline Lund model, as time evolves is gradually converted to potential energy stored in the string. In the baseline model, all of the kinetic energy is available to be converted to potential energy; they are in 1-1 correspondence; therefore there is no need to book keep them separately. Technically, in the Lund model, string systems are therefore represented entirely in terms of their momentum-space boundary conditions; the energies and momenta of their endpoints. In our repulsion model, however, a portion of the original longitudinal momentum is converted to transverse momentum; that is, a certain fraction of the CM energy *remains kinetic* and is therefore not available for particle production; the total produced particle multiplicity must therefore decrease.

However, if the Referee is referring to the notion of reducing the effective mass of the strange quark, we would like to clarify the following point: strangeness enhancement is not a part of this repulsion model. We believe that one could unify the description of strangeness enhancement (in the manner that the rope model works) and our repulsion, such that the two aspects of collectivity derive from the same root. However, for this work, we solely focus on the repulsion component. We also note that, in the rope model, it is not that the strange quark’s mass decreases, so much as the string tension increases. This leads to an effective decrease of the strange quark’s mass - relative to the string tension. That is, the effective quark masses are regarded as being unchanged; what is changing is the effective string tension, and the higher that gets, for constant constituent quark masses, the closer the u:d:s ratios get to 1:1:1. We plan to incorporate this aspect in future work.

With regards to the second point, since we are working with a Monte Carlo event generator, we cannot a priori know anything about the multiplicity of the final state. However, we do know that a single Lund string will, on average, produce one hadron per unit of rapidity. Accordingly, in the Lund string model, high multiplicity events occur when there are many long strings. Since we have only considered a pair of strings, it may be unwise to extrapolate these results directly to situations involving many strings. However, we believe it is safe to say that one would be able to use lower values for the repulsion constant for the same increases/decreases on observables such as c_2{2}, since each string will be interacting with several others.

With regards to the third point, we have now included a new paragraph in the introduction with many more references to the vast amount of research conducted by others from different approaches, which reads as follows:
'Other approaches to studying and modeling azimuthal correlation sin proton-proton collision environments include the String Percolation model [46–50], or interference effects from multiple parton interactions [51] or from the BFKL parton shower evolution process [52]. Recent work has investigated the initial state geometry of the collision and the resultant effect on azimuthal anisotropy [53–55]. The Colour Glass Condesate (CGC), a successful framework for describing the collision environment in heavy-ion collisions, has also been applied to proton-proton collisions [56, 57]. Kinetic transport theory has also been used to study the potential source of angular correlations [58]. A review of collectivity in small systems can be found in e.g. [59, 60].'

---

## Round 1 · Referee Report · Anonymous (Referee 3) · 2020-3-13

Report

The authors present steps towards a modification of the string hadronization with the attempt of modeling collective phenomena in hadronic collisions, specifically high-multiplicity proton proton collisions. The basic motivation of repulsion between flux tubes has already been addressed in the shoving model several years ago, by Bierlich and collaborators.

The authors study, in a well-written and understandable paper, the string repulsion dynamics between two strings, though they do not consider the steps required to include gluon radiation (i.e. kinks on the strings), and how a realistic model would have been built. As such, the work presented might be regarded as incomplete. As a first step to developing an alternative or new model, the work however deserves publication in order to document and foster the scientific discussion on this subject, pending small modifications: The authors' selection of literature on color reconnection based on color charge arguments or string junctions seems to be rather biased, and could be extended to reflect similar approaches performed by other groups. Also it should be avoided to generate an impression that the work presented in the manuscript (as with all phenomenological models) might be anywhere near being calculable from first principles QCD.
  • validity: -
  • significance: -
  • originality: -
  • clarity: -
  • formatting: -
  • grammar: -

Author:  Cody B Duncan  on 2020-04-22  [id 805]

(in reply to Report 3 on 2020-03-13)

We thank the Referee for recognizing the value of our manuscript as a prototype for a string repulsion model that is an alternative to the shoving model by Bierlich et al.
With regards to their two major points, we have replaced references to "QCD strings" with the more appropriate term "Lund strings".
Similarly, we have included more references to work conducted by other groups, and in particular, other approaches to modelling the ridge phenomenon in proton-proton collision environments. The added paragraph reads as follows:
'Other approaches to studying and modeling azimuthal correlation sin proton-proton collision environments include the String Percolation model [46–50], or interference effects from multiple parton interactions [51] or from the BFKL parton shower evolution process [52]. Recent work has investigated the initial state geometry of the collision and the resultant effect on azimuthal anisotropy [53–55]. The Colour Glass Condesate (CGC), a successful framework for describing the collision environment in heavy-ion collisions, has also been applied to proton-proton collisions [56, 57]. Kinetic transport theory has also been used to study the potential source of angular correlations [58]. A review of collectivity in small systems can be found in e.g. [59, 60].'

---

## Round 3 · Referee Report · Anonymous (Referee 3) · 2020-4-29

Report

The authors have been addressing my main criticism, so I am happy to recommend the manuscript for publication.

---

## Round 3 · Referee Report · Anonymous (Referee 2) · 2020-5-7

Strengths

The paper deals with the implementation of two point cumulant in the Lund model. The authors show that the two point cumulant is naturally implemented if one assumes the string repulsion proportional to rapidity overlap between strings. They show that there exists the value interval of their parameter c_R where the
cumulant has a right order of magnitude, although it is not clear how much physical this is since only the two string configurations are accounted for
In a new version the paper became much more understandable
and easy to read

Weaknesses

the weakness of the paper is ad-hoc ansats for repulsion
which does not seem to follow from anywhere. the major check of such ansats may be the study of multi particle correlators and check
for their collectivity (they change signs)-this was not achieved in
any of CGC based model, although derived in ref. [51].
I do not think this weakness should prevent however the publication of the article, since Lund model is itself rather phenomenological model, and all its successful modification should be welcomed.

Report

I believe the author took into account all my remarks and the paper should be published.

---

## Round 3 · List of Changes

1. Replaced all instances of the phrase "QCD strings" with the more appropriate term "Lund strings", since we are working with the Lund string model
  2. Added an introductory paragraph with references to a number of different research approaches.

---

## Editorial Decision

published